# DO-AutoEncoder: Learning and Intervening Bivariate Causal Mechanisms in Images

## Abstract

Some fundamental limitations of deep learning have been exposed such as lacking generalizability and being vunerable to adversarial attack. Instead, researchers realize that causation is much more stable than association relationship in data. In this paper, we propose a new framework called do-calculus AutoEncoder(DO-AE) for deep representation learning that fully capture bivariate causal relationship in the images which allows us to intervene in images generation process. DO-AE consists of two key ingredients: causal relationship mining in images and intervention-enabling deep causal structured representation learning. The goal here is to learn deep representations that correspond to the concepts in the physical world as well as their causal structure. To verify the proposed method, we create a dataset named `PHY2D`, which contains abstract graphic description in accordance with the laws of physics. Our experiments demonstrate our method is able to correctly identify the bivariate causal relationship between concepts in images and the representation learned enables a do-calculus manipulation to images, which generates artificial images that might possibly break the physical law depending on where we intervene the causal system.

## 1 Introduction

Recent breakthrough of deep learning (LeCun et al., 2015; Goodfellow et al., 2016) has significantly advanced the development of Artificial Intelligence (AI). However, statistical prediction and inference models, no matter traditional machine learning models such as random forest, support vector machines or deep models have their limitations such as lacking generalizability, being vulnerable to adversarial attacks as well as lacking interpretability (Zhang et al., 2016; Papernot et al., 2016). All these limitations prevent AI from wide applications to high stake scenarios including healthcare, autonomous driving (Chen et al., 2015) etc. Tremendous amount of efforts from the AI community has been devoted to understanding and improving the robustness of deep models and researchers come to consensus that after training with huge amount of data, deep models exploit "superficial" association in data to make prediction. However, such kind of subtle association relationship is unstable across domains. For example, the association relationship between pixels related to "grassland" and the label "elephant" is real but it is rather weak. Deep models, trained with hundreds of thousands of natural images where elephants reside on grassland, exploit too much pixel level association of the image to the label. Unfortunately, such models fail to detect an elephant if the background of the image is changed to a living room. More and more evidences show that deep models exploit association instead of causation when making the prediction, rendering it weak in term of domain generalization and robustness against adversarial attacks. As a consequence, researchers are advocating bringing causality into the life-cycle of deep neural networks.

In Judea Pearl book "The book of why" (Pearl & Mackenzie, 2018), he gives ladder of causation: The lowest level is concerned with patterns of association in observed data. The next level focuses on intervention and the highest one is counterfactual inference. In order to realize real Machine Intelligence, we must replace reasoning by association with reasoning by causality. However, how to incorporate causality into current machine learning framework is very challenging and little work has been found yet until the most recently, researchers start to look at disentangled representation learning. In the deep learning community, researchers realized that a good representation enables better generalizability and might possibly enjoy better interpretability. Disentangled representation is believed to be a good representation and become a research focus in a past few years (Hsu et al.,

2017; Narayanaswamy et al., 2017; Locatello et al., 2018). The idea behind disentangled representation learning is that we prefer the learned representations as independent as possible such that they correspond to the independent generative factors of the image, e.g. the illumination, azimuth, elevation etc for 3D graphics. The advantage of disentangled representation is that it offers better explainablity and it is claimed that deep models with disentangled representations enjoy better generalization performance for downstream tasks.

Although significant progress has been achieved in the area of deep representation learning in the past few years, most of existing work assume that the representations or the latent codes are statistically independent and they are trained by adding a regularization term of mutual information between latent codes to the loss function (Chen et al., 2016). It is of great interest and has shown advantage over entangled representation, but forcing the latent codes to be independent might limit itself both in term of theoretical understanding and practical applications. In various scenarios, we are rather interested in learning latent representations that correspond to the causally related concepts. For example, given an image consisting concepts like tree, sunshine and shadow, an idea representation learning algorithm is expected to yield latent codes that could each corresponds to the above three concepts. Obviously, these three concepts are by no means independent, i.e. the concept of tree and that of sunshine together causes the concept of shadow. If we are able to extract latent codes with their causal structure, we are able to intervene the whole system, e.g. intervene the latent code of shadow such that we can generate images with the shadow removed. This could be potentially of great value in many applications such as autonomous driving.

In this paper, we propose a framework called DO-AutoEncoder (DO-AE) to learn causally related concepts in images, which is achieved by firstly discovering the causal relationship of concepts in the images by comparing complexities of generative models ($\mathcal{G}(\text{cause})$, $\mathcal{G}(\text{effect}|\text{cause})$) in either hypothetic directions and the generative models are realized by AutoEncoder structures. The latent codes are then used to reconstruct the original input images and the whole AutoEncoder structure is trained by minimizing the reconstruction error. The framework mimics the whole data generating mechanism, i.e. the joint distribution $P(\text{cause}, \text{effect})$ is factorized as $P(\text{cause})P(\text{effect}|\text{cause})$ according to the causal direction. As Figure 10 shows, the light is projected from the vertical direction upside down, yielding the time-varying shadow of a swinging pendulum. So the causal direction is from angle (of pendulum) to length (of shadow). We aim at developing deep representation learning model and algorithm to discover such causally related concepts from a series of images describing the law of physics. In addition to the pendulum data, in this paper we create a series of dataset named PHY2D, the collection of images of abstract graphic description of physics laws.

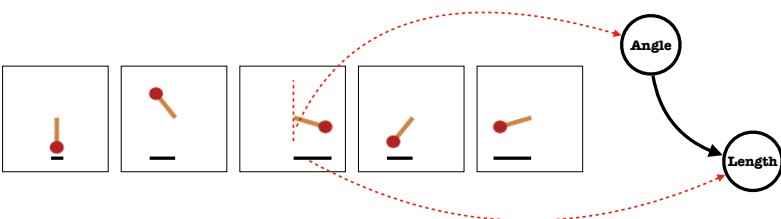

Figure 1: The laws of physics can be expressed by functions, and the independent variables and dependent variables in the function can be abstracted to the pixels in the image. These pixels therefore have a causal direction. Finally, the real physical world can be abstracted into a causal diagram.

## 2 RELATED WORK

People have studied causality for a long time and related work can be found in philosophical publications, religious scriptures and science publications. Ancient Greek famous thinker Aristotle proposed "Four causes" which includes the final cause, the material cause, the formal cause and the efficient cause. He believes that all natural phenomena follow the causal law. Scottish philosopher David Hume recognizes cause-effect relation as both a philosophical relation and a natural relation, and he argued that this relation is pivotal in reasoning (Hume, 2003). (Harari, 2014) provide that human ancestor's causal imagination is the basis for mutual communication. The instinct of human

beings to explore the cause and effect of things makes human beings more intelligent, and causal thinking is the foundation of the development of science and technology. Causality is a bridge between science and philosophy.

In science, causality can be found in several fields such as economics (Moneta et al., 2013), neuroscience (Ramsey et al., 2014), environment science (Sugihara et al., 2012) and especially machine learning. Pearl shows that machine learning without causal inference will encounter Yule-Simpsons Paradox, which means the model can get two completely contradicting conclusions with the same dataset. Recent efforts from machine learning community try to link causality and ML methods. To name a few, (Schölkopf et al., 2012) argues that semi-supervised learning only works in anti-casual direction. (Bengio et al., 2019) propose a meta-learning framework to determine the causal relationship between two observed variables. The correct modeling according to the causal direction usually leads to faster adaptation to shifted data distribution due to interventions to the system.

Using machine learning for analyzing physics is a hot topic (Watters et al., 2017; Fragkiadaki et al., 2015; Janner et al., 2018). (Bakhtin et al., 2019) provide a 2D physical classical mechanical puzzle dataset to encourage more useful models for physics. In interdisciplinary field of computer vision and causality, (Lebeda et al., 2015) provide statistical tools based on transfer entropy to explore the causality relationship between the camera motion and the object motion to predict object location. The prediction accuracy is increased over non-causal predictor. (Lopez-Paz et al., 2017) focus on establishing the existence of causality based on observable statistical signal carried in image datasets, by using causal discovery technique Neural Causation Coefficient(NCC). These recent advances show that causal inference in images is gaining increasing amount of attention from the AI community.

## 3 METHOD

As mentioned in Sec 1, we aim at representation learning that correspond to the concepts as well as their causal structure in images. To achieve this target, the first step is to discover the causal relationship between the concepts in images, i.e. we need to identify which one is the cause or effect. In this paper, we follow the Independent Mechanism principle proposed by (Janzing & Scholkopf, 2010). Secondly, we need to learn the generating mechanisms of the cause and the generating mechanism mapping the cause to the effect. We propose to use AutoEncoder as the generative model.

Given observations of two variables $X$ and $Y$, distinguishing the cause from effect is a challenging problem. There is a postulate named Independent Mechanism principle (Janzing & Scholkopf, 2010) stating that the joint distribution factorization has asymmetric statistical behaviors in causal versus anticausal directions. Specifically, decomposition of the joint distribution according to the causal direction is natural and simpler than the anti-causal direction. This principle can also be explained by the famous Occam's Razor principle. If there are multiple models that can explain the data equally well, we prefer the simplest one.

The definition of simplicity varies in different methods. For example, (Janzing & Scholkopf, 2010) argues that the Kolmogorov complexity of conditional and marginal distributions is smaller in causal direction than that in anticausal direction. It can be written as the following inequation:

$$\mathcal{K}(P_X) + \mathcal{K}(P_{Y|X}) \leq \mathcal{K}(P_Y) + \mathcal{K}(P_{X|Y}), \tag{1}$$

where $\mathcal{K}(\cdot)$ is the Kolmogorov complexity. However, the Kolmogorov complexity is not computable, and thus researchers proposed other statistical methods including Mimimal Discription Length (MDL) or Reproducing Kernel Hilbert Space (RKHS) norms (Sun et al., 2007) as approximation. Recently, (Bengio et al., 2019) showed that the joint distribution decomposed according to the causal direction is much easier to learn and is able to adapt much faster to the distributional shift in data than the anti-causal direction. Inspired by this work, in this paper we propose to learn the causal direction of concepts in images according to model complexities.

Firstly, we introduce how to identify the causal direction according to the loss after model training. Denote by $A$, $B$ the ground truth hypothetic cause and effect concepts in the images, e.g. in the pendulum case, if we assume that the pendulum causes the shadow, then $A$ corresponds to the

concept of pendulum and $B$ corresponds to the concept of shadow. We further denote by $\hat{A}$ and $\hat{B}$ the corresponding generated concepts by two generative models $f(\text{enc}; \boldsymbol{\theta})$ and $g(\hat{A}; \boldsymbol{\eta})$, where $\text{enc}$ is the original input image and $\boldsymbol{\theta}$ and $\boldsymbol{\eta}$ are model parameters. In this paper, we use AutoEncoder the generative functions.

The likelihood of the generative concepts is defined as follows:

$$P_{\mathcal{G}}(\hat{A}, \hat{B}) := \exp(-\|A - \hat{A}\|_2) \exp(-\|B - \hat{B}\|_2) \tag{2}$$

Two generative models, i.e. the model generating the cause and the model mapping the cause to the effect are then trained by minimizing the negative log-likelihood as follows:

$$\boldsymbol{\theta}^*, \boldsymbol{\eta}^* = \arg\min \mathbb{E}[\|A - f(\text{enc}, \boldsymbol{\theta})\|_2] + \mathbb{E}[\|B - g(\hat{A}; \boldsymbol{\eta})\|_2]$$

Note that $\mathbb{E}[\|B - g(\hat{A}; \boldsymbol{\eta})\|_2] = \mathbb{E}[\|B - g(A; \boldsymbol{\eta}) + g(A; \boldsymbol{\eta}) - g(\hat{A}; \boldsymbol{\eta})\|_2] \leq \mathbb{E}[\|B - g(A; \boldsymbol{\eta})\|_2] + \mathbb{E}[\|g(A; \boldsymbol{\eta}) - g(\hat{A}; \boldsymbol{\eta})\|_2] \leq \mathbb{E}[\|B - g(A; \boldsymbol{\eta})\|_2] + L\mathbb{E}[\|A - \hat{A}\|_2]$, if we assume that $g$ is Lipschitz continuous and we denote by $L$ the Lipschitz constant.

We then can get the lower bound of the log-likelihood, i.e.

$$\mathcal{L}_b = -(1 + L)\mathbb{E}[\|A - f(\text{enc}, \boldsymbol{\theta})\|_2] - \mathbb{E}[\|B - g(A; \boldsymbol{\eta})\|_2] \tag{3}$$

However, this still requires choosing $L$ which might be unknown. We further relax the loss function to be

$$\mathcal{L}_{A \to B} = -(1 + L)\mathbb{E}[\|A - f(\text{enc}, \boldsymbol{\theta})\|_2] - (1 + L)\mathbb{E}[\|B - g(A; \boldsymbol{\eta})\|_2] \tag{4}$$

To identify the causal direction, we only need to compare which direction gives lower loss, given the same model complexity, i.e.

$$\begin{cases} A \to B, \text{ if } \mathcal{L}_{A \to B} > \mathcal{L}_{B \to A}; \\ B \to A, \text{ if } \mathcal{L}_{B \to A} < \mathcal{L}_{A \to B}; \\ \texttt{NO conclusion, otherwise.} \end{cases} \tag{5}$$

Our whole pipeline is in Figure 2. The whole DO-AE inputs an image and then tries to reconstruct it at the output. Instead of using a single AutoEncoder, we propose a modularized one by decomposing the whole system into two sub-modules, and each is an AutoEncoder. The first module aims at generating the hypothetic cause, followed by a second one to generate the hypothetic effect from the hypothetic cause. By controlling the model capacity, the likelihood are then used to decide which direction is more plausible to be the causal direction. We prefer the direction with higher likelihood according to the Independent Mechanism principle. One should be noted that we cannot use a model with too rich flexibility otherwise the aforementioned method might not work as it could be the case that with very expressive models, the likelihoods of both hypothetic direction are almost equally good and thus not distinguishable. We verify this conjecture in the experiment section. Note that in experiments we simply set $L = 0$ since its value would not have an impact on the inference rule.

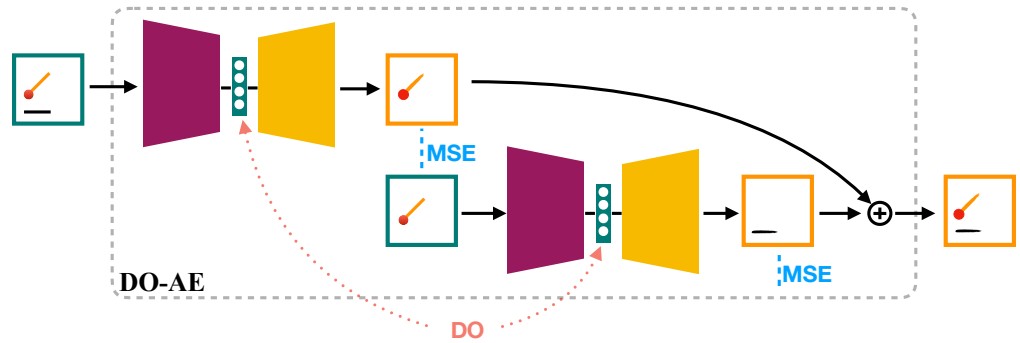

Figure 2: DO-AE model structure. The original images and its segmented parts are outlined in teal and the generated images are outlined in orange. This figure illustrates the true direction pendulum → shadow, and the same architecture applies to the opposite direction. We extract the latent code to intervene the whole system. Doing cause latent code does not affect the generation and doing effect code can generate images which break the physical law.

## 4 PHY2D: UNDERSTANDING PHYSICAL LAWS IN IMAGES

We create a dataset named PHY2D to verify the effectivness of our proposed approach. The dataset contains seven category of abstract graphic descriptions in accordance with the laws of physics. The detailed descriptions are shown in Table 1. Each category contains 5000 figures.

In data BEVEL, the height of the ball generates the gravity potential energy, and gravity potential energy converts to kinetic energy based on energy conversion. For the elements in the figure, the red circles height and the wooden board shape cause the blue circles' track and the leftmost blue circle's position. In data PENDULUM, the light is projected upside down, because light propagates straightly, the swinging pendulum blocks light, creating shadow with varying length. Other dataset can be explained as follows: (1) BUOYANCY: The volume of ball causes the water level; (2) SPRING: The mass of ball causes the spring's length; (3) BOWL: The initial position of the red ball causes the rolling track; (4) CONVEX: The width of incident light causes refracted light direction. (5) FLOW: Water column height causes the trajectory of water flow. The example pictures of the whole dataset is showed in Figure 3 and it contains $7 * 5000 * 3$ figures.

Table 1: The PHY2D dataset detailed descriptions.

| Order | Name | Physical law | Description |
|-------|------|--------------|-------------|
| 1 | BEVEL | Energy conversion | A small ball rolls off from a frictional bevel. The red ball stands for the initial state and the motion track is shown as blue balls. |
| 2 | PENDULUM | Rectilinear propagation | A rotating red and yellow colored pendulum and its shadow. |
| 3 | BUOYANCY | Buoyancy force | Balls with different sizes are placed in a measuring cylinder filled with water. |
| 4 | SPRING | Elastic force | Balls of different weights hung from the lower end of the spring. |
| 5 | BOWL | Energy conservation | A ball is released from the smooth inner wall of the bowl. The blue circles stand for the rolling track. |
| 6 | CONVEX | Refraction law | Parallel light rays converge at the focus of the convex lens. |
| 7 | FLOW | Pressure | Water is flowing out from the hole on the cylinder filled with water. |

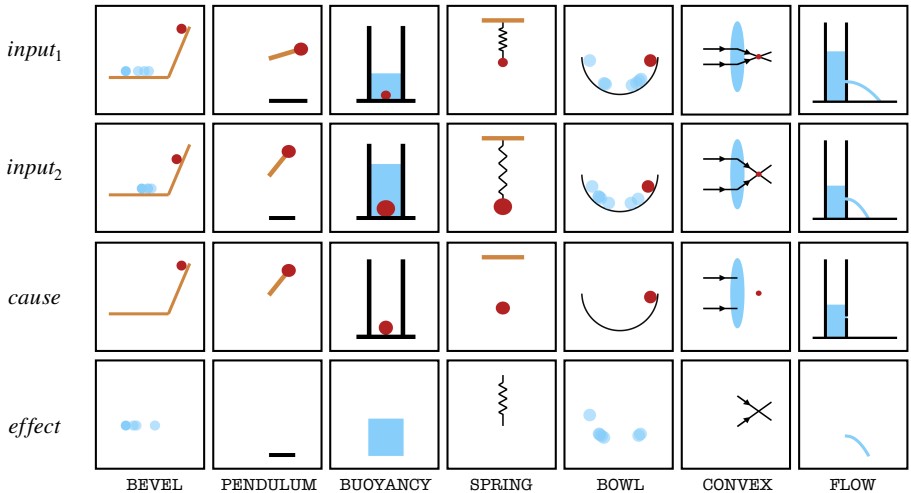

Figure 3: `PHY2D` dataset.

## 5 EXPERIMENTS

In this section, we present our experimental results. We firstly give a detailed description of the network architectures used in our paper. Then, we analyze the results of the experiments on different datasets. In encoder and decoder framework, we use traditional convolution neural network(CNN). We design two kinds of network structures: simple structure and flexible structure. The network architectures are shown in Table 2[1]. The whole experiments are conducted with the following settings:

- Operating system: Ubuntu 16.04.

- Software version: Python 3.5.2; Pytorch 1.0.1; Numpy 1.17.0.

- GPU: NVIDIA TESLA V100.

Table 2: Network architectures for causality discovery.

| Simple structure | Flexible structure |
| --- | --- |
| Input $96 \times 96$ 3-channel Color image | |
| $4 \times 4$ conv. 64 lRELU. stride 2. | $4 \times 4$ conv. 64 lRELU. stride 2. |
| $4 \times 4$ conv. 100 lRELU. stride 2. batchnorm | $4 \times 4$ conv. 128 lRELU. stride 2. batchnorm |
| $4 \times 4$ upconv. 64 lRELU. stride 2. batchnorm | $4 \times 4$ conv. 256 lRELU. stride 2. batchnorm |
| $4 \times 4$ upconv. 3 lRELU. stride 2. | $4 \times 4$ conv. 512 lRELU. stride 2. batchnorm |
| | $4 \times 4$ conv. 1024 lRELU. stride 2. batchnorm |
| | $4 \times 4$ conv. 100 lRELU. stride 2. batchnorm |
| | $4 \times 4$ upconv. 1024 lRELU. stride 2. batchnorm |
| | $4 \times 4$ upconv. 512 lRELU. stride 2. batchnorm |
| | $4 \times 4$ upconv. 256 lRELU. stride 2. batchnorm |
| | $4 \times 4$ upconv. 128 lRELU. stride 2. batchnorm |
| | $4 \times 4$ upconv. 64 lRELU. stride 2. batchnorm |
| | $4 \times 4$ upconv. 3 lRELU. stride 2. |
| Output $96 \times 96$ 3-channel Color image | |

---

[1]For the dataset `PENDULUM`, we use a specific network (different from simple or flexible ones). The network architecture used on `PENDULUM` is in Table 3 in appendix A.

Figure 4 shows the likelihood curves on data FLOW using our two architectures. Due to the powerful fitting capability of neural networks, if the causal mechanism of the image is relatively simple, the curves of two hypothetic directions are not distinguishable. On the contrary, a simple neural network architecture with capacity control is able to make the asymmetry. One can notice that even with a simple neural network architecture, a generative model according to the causal direction yields a larger likelihood than the anti-causal direction. This shows that generating the image according to the true causal direction may be much simpler and more straightforward.

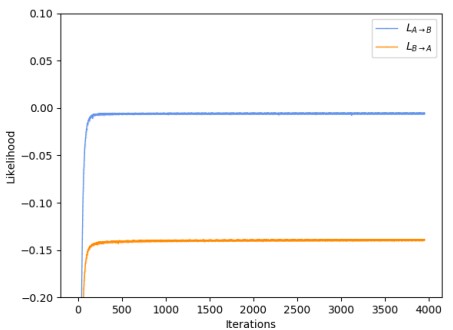
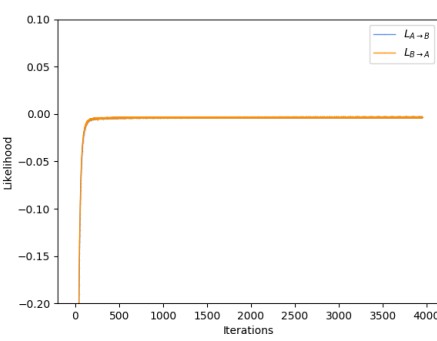

(a) Likelihood curves of simple network structure.  (b) Likelihood curves of flexible network structure.

Figure 4: Likelihood curves on FLOW.

We plot two kinds of experimental results on different datasets: the images generated by the AEs in first 5 training epochs (as illustrated in Fig.5) and curves of model likelihood during training process (as illustrated in Fig.6).

From Fig.5, we find that the images generated in the causal direction is in general of better quality than those in the anti-causal direction, which aligns with our initial conjecture and justifies the effectiveness of our proposed method.

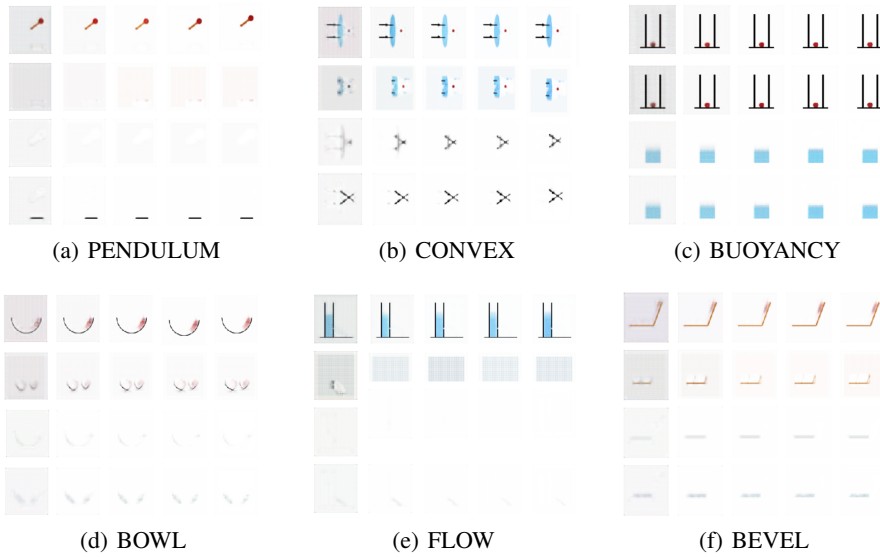

(a) PENDULUM          (b) CONVEX          (c) BUOYANCY

(d) BOWL          (e) FLOW          (f) BEVEL

Figure 5: The cause and effect images generated by AEs during the first 5 training epochs in two hypothetic directions. Figures on the first row illustrate the generated hypothetic cause images in the causal direction $A \rightarrow B$. The second row shows the generated hypothetic cause images in anti-causal direction $B \rightarrow A$. The third and fourth rows show the corresponding hypothetic effect images respectively.

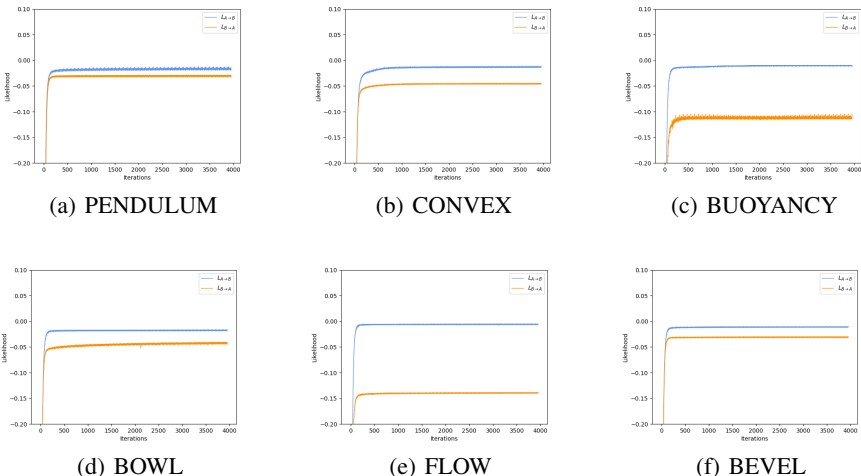

| | | |
|---|---|---|
| (a) PENDULUM | (b) CONVEX | (c) BUOYANCY |
| (d) BOWL | (e) FLOW | (f) BEVEL |

Figure 6: Likelihood curves of DO-AE models on 6 datasets with simple network architecture.

## 5.1 RESULT ANALYSIS ON PENDULUM

We here specifically analyze 2 datasets. First, we show experimental results when AE is designed using the flexible network structure on dataset PENDULUM in Figure 7. Figure 7(a) shows the images generated by the AEs during first 5 training epochs. Figure 7(b) shows likelihood curves of models in $\mathcal{L}_{A \rightarrow B}$ and $\mathcal{L}_{B \rightarrow A}$ pathways. In $A \rightarrow B$ pathway, we find that cause-to-effect generation is simpler. The quality of effect image is acceptable. However, in $B \rightarrow A$ pathway, the cause images generated from the effect image include two pendulums in one picture. This is because the function corresponding to the causal mechanism is not invertible. Under this scenario, using the cause image to generate the effect image is better than using the effect image to generate the cause image. Another observation is that the likelihood curves in Figure 7(b) $A \rightarrow B$ way converges to zero faster. In $B \rightarrow A$ way, as the number of iterations increases, the curve begins to fluctuate, indicating that the effect images can not stably reason about the cause image.

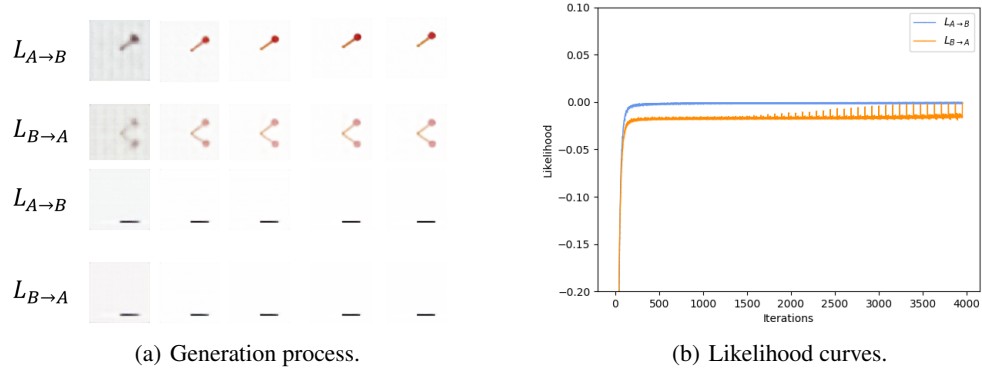

| (a) Generation process. | (b) Likelihood curves. |
|---|---|

Figure 7: Causal discovery results with flexible network architecture on PENDULUM.

## 5.2 RESULT ANALYSIS ON SPRING

We then analyze the dataset SPRING. We observe that both simple and flexible networks don't work on this dataset. We show the results (AE being the flexible network) in Figure 8. The generated figures in $A \rightarrow B$ pathway and $B \rightarrow A$ pathway are not very different. The likelihood curves show certain degree of entanglement. This indicates that DO-AE model cannot find the causal direction

between spring length and ball weights. However, this doesn't mean that our model lacks the ability of causality discovery. We try to explain this phenomenon by rethinking the dataset. Newton's third law of motion said for every action has an equal and opposite reaction. So the gravity generated by small ball causes the spring to undergo an elongated deformation. Meanwhile, the spring in the stretched state generates an elastic force, pulling the ball up. So, the spring length can also cause the position of ball. It is possible the case that DO-AE model could not conclude when it encounters a bidirectional causal scenario.

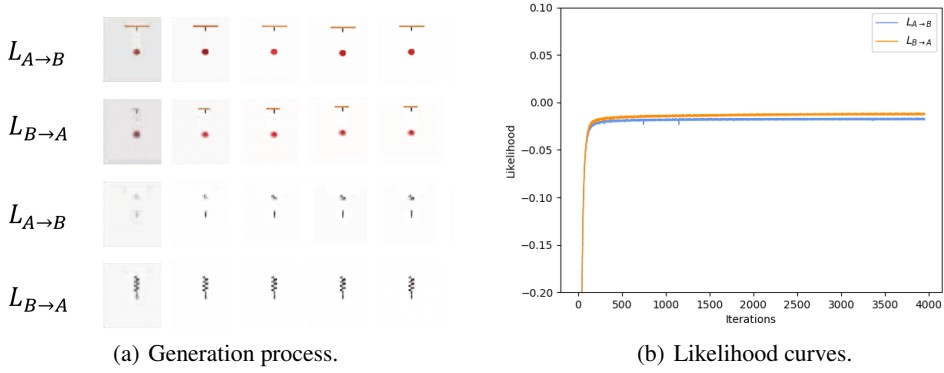

(a) Generation process.    (b) Likelihood curves.

Figure 8: Causal discovery results with flexible network architecture on `SPRING`.

### 5.3 Intervening the Image Generation Process

After the model training process for causal discovery, we can choose the causal pathway as our generative model and intervene the image generation process to achieve do-calculus. DO-AE model outputs two latent codes. The second latent code contains the information of the causal mechanisms. By fixing the first or second latent code in the whole generating process, we intervene the whole model and the system outputs some counterfactual images. We show some images generated by DO-AEs trained on `PENDULUM` in Figure 9.

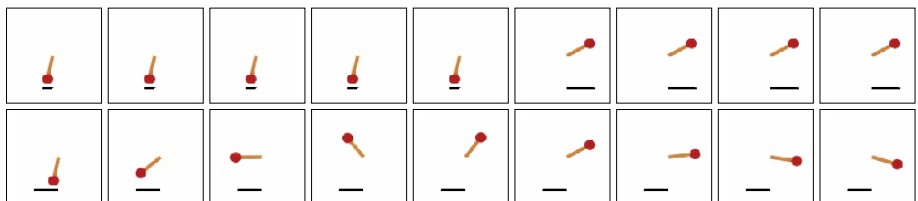

Figure 9: Do-calculus result. In the first row, we do the cause image to show the result image of $P((\text{cause}, \text{effect})|do(\text{cause} = A_0))$ (first 5 pictures) and $P((\text{cause}, \text{effect})|do(\text{cause} = A_1))$ (last 4 pictures), and the second row shows $P((\text{cause}, \text{effect})|do(\text{effect} = B_0))$.

## 6 Conclusion

In this paper, we propose a deep learning framework DO-AE to discover bivariate causal relationships in images. Experiments suggest that DO-AE can learn the causal relationship between objects. We perform interventions on the generative model to produce counterfactual images that break the physical law. In the future, we hope to provide an approach for mining multivariate causal relationships, and provide a causal way to solve the combinatorial explosion problem.

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

## A  APPENDIX

The network structure used in Sec. 5. When we use our simple network structure, the likelihood curve is shown below.

Table 3: Network architectures on Data `PENDULUM`.

| **Network structure** |
| --- |
| Input $96 \times 96$ 3-channel Color image |
| $4 \times 4$ conv. 64 lRELU. stride 2. |
| $4 \times 4$ conv. 128 lRELU. stride 2. batchnorm |
| $4 \times 4$ conv. 100 lRELU. stride 2. batchnorm |
| $4 \times 4$ upconv. 128 lRELU. stride 2. batchnorm |
| $4 \times 4$ upconv. 64 lRELU. stride 2. batchnorm |
| $4 \times 4$ upconv. 3 lRELU. stride 2. |
| Output $96 \times 96$ 3-channel Color image |

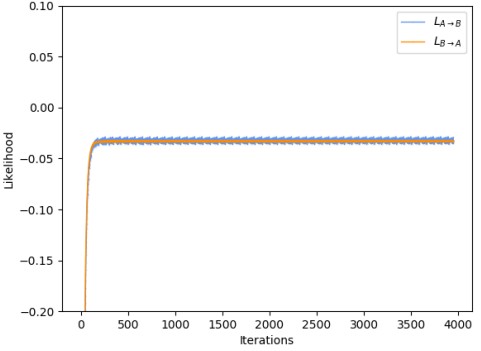

Figure 10: Likelihood curves on `PENDULUM`

