# OpenReview forum: "DO-AutoEncoder: Learning and Intervening Bivariate Causal Mechanisms in Images"
_ICLR.cc/2020/Conference — Reject_

### Official Review · AnonReviewer2 · 2019-10-22
**Official Blind Review #2**

**Rating:** 1

**Review:**

This paper presented an image data that are generated from two variables using some physics law. It also proposed a model to identify the causal relationship between the two variables using the image dataset. The method, in general, utilize the general idea that the causal direction is easier for the model to describe than the anti-causal direction. So the image is fad into a VAE based model in two different ways. The one with lower loses represents the correct causal direction.

Pros:
1. Causal discovery is, in general, an interesting problem and causal discovery based on representation learning are of great importance.
2. The dataset presented can be used for generic causal discovery evaluation which can be useful for the community.

Cons and other details:
1. The method assumes that A and B are known and given which is very unrealistic in natural images. Also with this assumption, the problem is not much different from causal discovery from measurement data rather than image data.
2. Based on the previous point, the method, in general, does not match the motivation in the introduction where a causal representation needs to be learned as the images are already separated into different components.
3. The method cannot be scaled to more than two variables even with all components given as it requires exponentially many trials of the method. This setting is not so interesting anymore with image input.
4. There is much-related work with causality and representation learning also causality with NN or VAE. None of these related work has been discussed.  for example Leon Bottou https://arxiv.org/pdf/1907.02893.pdf; Many works from Mingming Gong etc
5. The math is not very rigorous in general. For example, Eq(2) s a valid-loss but not likelihood. Also, the work did not say what likelihood under what distribution. This is propositional to Gaussian likelihood which may work fine in practice but the math presentation is not rigorous.
6. For the method (see figure 2), I did not see why the first part needs to be there as the second part takes the ground truth A as input. Using only the second part of the model which tries to see whether A->B is easier or B->A is easier is sufficient for the aim of identifying the relationship between given A and B.
7. The dataset may be more useful to the causality community if it is released as a simulator rather than the images.


**Experience Assessment:**

I have published one or two papers in this area.

**Review Assessment: Checking Correctness Of Derivations And Theory:**

I assessed the sensibility of the derivations and theory.

**Review Assessment: Checking Correctness Of Experiments:**

I assessed the sensibility of the experiments.

**Review Assessment: Thoroughness In Paper Reading:**

I read the paper thoroughly.

---

> ### Author Response · Authors · 2019-11-15
> **Response to reviewer #2**
>
> We are grateful for R2's constructive suggestions and we believe they could greatly improve our paper.
>
> -As your proposition said, though for our dataset the two variables are given, it is quite different from causal discovery from measurement data. Our model fits for high dimensional visual data, which is a main contribution of this work.
>
> -We separate the image into two part for two reasons:
>   1.Reduce the mutual interference between variables.
>   2.Our paper is mainly based on the following assumption: The Kolmogorov complexity of conditional and marginal distributions is smaller in causal direction than that in anti-causal direction. In Figure 2, the part I of DO-AE is to estimate K(P_x), the part II is to estimate K(P_{y|x}). The outputs of these two part make up the whole image, we want to intervene in whole images generation process, so two parts of the DO-AE are indispensable.
>
> -We will access related work you recommend，and cite the related work about causality with VAE.
>
> Thank you again for your detail comments.

---

### Official Review · AnonReviewer3 · 2019-10-22
**Official Blind Review #3**

**Rating:** 1

**Review:**

In summary: This paper is not ready for publication. The paper contains some potentially interesting ideas, but the presentation quality is not sufficient for publication. The paper should be substantially improved before re-submission.

Strengths:
- Causality is an important and established research area, and papers on this topic would be timely.
- Paper contains some interesting ideas to integrate causality into an auto-encoder (but see weaknesses below)
- Paper proposes a new dataset for evaluating causal mechanisms (but the approach is not evaluated)

Weaknesses:
- The quality of the writing is inappropriate for a scientific venue. Language throughout the paper is loose, eg "physics is a hot topic" or "People have studied causality for a long time" or "Causality is a bridge between science and philosophy" The paper should be re-written so that it is precise and clear.
- The technical approach has several typos and lacks discussion of the approach. Instead, several high-level statements are made, with long equations. This makes appreciating the contribution of the paper difficult.
- The dataset is potentially interesting, but it is artificial. A much more exciting dataset would be realistic data.
- The experiments only evaluate the likelihood, but it is not clear whether this is on a training or testing set.


**Experience Assessment:**

I have read many papers in this area.

**Review Assessment: Checking Correctness Of Derivations And Theory:**

I assessed the sensibility of the derivations and theory.

**Review Assessment: Checking Correctness Of Experiments:**

I assessed the sensibility of the experiments.

**Review Assessment: Thoroughness In Paper Reading:**

I read the paper at least twice and used my best judgement in assessing the paper.

---

> ### Author Response · Authors · 2019-11-15
> **Response to reviewer #3**
>
> We thank reviewer #3 for his review.
>
> -We will enhance the writing ability to make the narrative of the paper more professional and rigorous.
>
> -There are a number of variables and complicated causal graph in natural images, we want to use the artificial data to conduct a preliminary exploration. Our next work will focus on discovering  causal relationship in real world.
>
> Thank you again for your time sincerely.

---

### Official Review · AnonReviewer1 · 2019-10-23
**Official Blind Review #1**

**Rating:** 6

**Review:**

Thank you for your submission.

- What is the specific question/problem tackled by the paper?

The paper proposes a VAE architecture to learn causal relations and allow for interventions. The architecture requires knowledge of the causal graph, and the direction of the causal arrows are inferred by comparing the log-likelihoods of generated images. The architecture may also require knowledge that an arrow exists between two vertices. This relies on the principle that "low-capacity" neural networks can predict better along the causal arrows (with the cause as input and the effect as the output) than in the opposite direction (with the effect as input and the cause as the output).

The paper focuses on the graph (A, B) where one wants to understand whether A causes B, or B causes A. The paper also discusses intervening in this graph.

The paper uses a new dataset for evaluating the approach, based on simple Newtonian systems.

- Is the approach well motivated, including being well-placed in the literature?

I think the motivation is adequate, but the review of the literature glosses over related work (or the absence thereof) in predicting the direction of arrows in causal graphs. The comparison of the proposed dataset against existing ones is missing.

- Does the paper support the claims? This includes determining if results, whether theoretical or empirical, are correct and if they are scientifically rigorous.

The procedure for determining whether A causes B (or B causes A) is qualitative. The paper demonstrates that the performance gap between the correct and incorrect explanations is consistently distinguishable across multiple experiments.

Visual inspection of the generated images is also used for assessing the quality of the models.

Because the results are qualitative, the support for the claims is not as strong as it could be (with quantitative results).

- Summarize what the paper claims to do/contribute. Be positive and generous.

The paper has two main contributions:
* Evidence to the Independent Mechanism principle (in a setting different from Bengio et al.'s transfer setup).
* A new dataset for evaluating learning causal arrows (with accessible ground-truths).

I think these are interesting contributions.

- Is the paper clearly written?

The paper has a number of grammatical errors that should be fixed.

The explanation of how the latent interventions are made is important and should be included.

- Clearly state your decision (accept or reject) with one or two key reasons for this choice.

I vote for a weak accept.

- Provide supporting arguments for the reasons for the decision.

I trust that the writing issues will be addressed in due course, but I am also concerned about the fact that evaluations are qualitative. The qualitative results provide support for the contributions that could be strengthened.

The dataset is also an interesting contribution and it is a good idea to give it visibility. For this, though, it is important that the paper assess its strengths and limitations in comparison to alternative datasets.

- Provide additional feedback with the aim to improve the paper. Make it clear that these points are here to help, and not necessarily part of your decision assessment.

I am not convinced that mentioning Kolmogorov complexity is an efficient use of the space. I think the content could be improved by making the motivation section more concise and adding a few more experimental results or discussion.

Which discussions would be good to have? I think it should be noted that the intervention on effect should behave as demonstrated (creating implausible scenarios). Also some more development on the spring example: What is the right causal graph for it, and can the arrows in that graph be learned?

Quantitative results would also improve the paper. Maybe decide between A->B or A<-B based on a statistical test?

You give an example about elephant-grassland association. Please cite a source for that.

Suppose that both likelihoods for A->B and B<-A are about the same. How do you decide if your model is too rich, or if there's no relationship? (This is an important question to understand if the method requires knowledge that an (A,B) arrow exists or not.)

The panels in Figure 5 do not support the claim. The simple net gets better at the cause, but in some cases the rich representation does a better job at the effect.

I think the physics dataset is also a contribution, so its originality & impact should be discussed in comparison to related work. Why is this an adequate benchmark? How does it address limitations of other benchmarks that could be used to evaluate proposed solutions for the problem in question?

In summary, my suggestions for improving the paper are:
1) Make sure & demonstrate (by adequate discussion of related work) the originality of the contributions:
1.1) The method for detecting the direction of causal arrows.
1.2) The dataset as a benchmark for the problem being studied.
2) Report quantitative results across the dataset and maybe across multiple setups for each name/physical law, with good coverage. You may consider a test set where the parameters are within the sampling range of your training set, and also outside the sampling range (where success of the method would be even more interesting).

**Experience Assessment:**

I do not know much about this area.

**Review Assessment: Checking Correctness Of Derivations And Theory:**

N/A

**Review Assessment: Checking Correctness Of Experiments:**

I assessed the sensibility of the experiments.

**Review Assessment: Thoroughness In Paper Reading:**

I read the paper thoroughly.

---

> ### Author Response · Authors · 2019-11-15
> **Response to reviewer #1**
>
> We thank reviewer #1 for his comments of our paper.
>
> -First of all, we are sorry about the grammatical errors in this paper, we will fix them to increase the readability of the paper.
>
> -Our aim is to construct the causal graph from the given images. This kind of exploration is challenging, we provide a physics dataset to explore the possibilities that this problem can solve. Our model functionality is built on the presence of known arrows，and the causal graph reflected by the images in the dataset has an arrow. DO-AE focus on learning the direction of the arrow(No arrow situation is not within our consideration). By the way, the right causal graph for the spring example is: A <-> B.
>
> -We decide the net is rich or not by determining the quality of the generated images visually and intuitively. We agree that increasing the statistical experiments and setting quantitative estimate index could make the results more convincing. We will improve this part in next version.
>
> Thank you again for your feedback.

---

### Decision · Program_Chairs · 2019-12-19

**Decision:**

Reject

**Comment:**

The idea of integrating causality into an auto-encoder is interesting and very timely. While the reviewers find this paper to contain some interesting ideas, the technical contributions and mathematical rigor, scope of the method, and the presentation of results would need to be significantly improved in order for this work to reach the quality bar of ICLR.